# Information content of stream level class data for hydrological model calibration

Ilja van Meerveld[1], Marc Vis[1], Jan Seibert[1,2]

[1] Department of Geography, University of Zurich, Zurich, Switzerland
[2] Department of Earth Sciences, Uppsala University, Uppsala, Sweden

*Correspondence to*: Ilja van Meerveld (ilja.vanmeerveld@geo.uzh.ch)

**Abstract.** Citizen science can provide spatially distributed data over large areas, including hydrological data. Stream levels are easier to measure than streamflow and are likely observed more easily by citizen scientists. However, the challenge with crowd-based stream level data is that observations are taken at irregular time intervals and with a limited vertical resolution.
The latter is especially the case at sites where no staff gauge is available and relative stream levels are observed based on (in)visible features in the stream, such as rocks. In order to assess the potential value of crowd-based stream level observations for model calibration, we pretended that stream level observations were available at a limited vertical resolution by transferring streamflow data into stream level classes. A bucket-type hydrological model was calibrated with these hypothetical stream level class data and subsequently evaluated on the observed streamflow records. Our results indicate that stream level data can
result in good streamflow simulations, even with a reduced vertical resolution of the observations. Time series of only two stream level classes, e.g. above or below a rock in the stream, were already informative, especially when the class boundary was chosen towards the highest stream levels. There was some added value in using up to five stream level classes but there was hardly any improvement in model performance when using more level classes. These results are encouraging for citizen science projects and provide a basis for designing observation systems that collect data that are as informative as possible for
deriving model-based streamflow time series for previously ungauged basins.

*Keywords.* information content, stream levels, hydrological model calibration, citizen science, measurement resolution, ungauged catchments

## 1 Introduction

Streamflow data are crucial for water resources management decisions and the calibration of hydrological models. However, streamflow data are only available for a number of sites and gauging stations are not always installed at representative locations. There is, for instance, a lack of streamflow gauges in small headwater streams (Kirchner, 2006) and in developing countries (Mulligan, 2013). Although technological developments provide the possibility to expand the measurement network, the reality is that due to budget cuts, observation networks often shrink (Kundzewicz, 1997), rather than expand. Remote
sensing images can be used to estimate stream levels or streamflow, particularly for wide lowland rivers (Smith, 1997;

Milewski *et al.*, 2009; Pavelsky, 2014; Van Dijk *et al.*, 2016) but estimation of streamflow from satellite images is likely to remain problematic for small headwater streams.

Stream level data are easier to obtain than streamflow data because they do not require any information on the rating curve. Seibert and Vis (2016) tested if stream level data can be used to constrain a simple hydrological model. The results for ~600 catchments in the USA showed that level data can be surprisingly informative for hydrological model calibration. This applies especially for humid and wet catchments (defined as catchments where the annual precipitation is larger than the annual potential evapotranspiration), for which the median efficiency (Nash and Sutcliffe, 1970) of models calibrated with stream level data were generally only about 0.10-0.15 units below those of models calibrated with streamflow data, and for all but one catchment the difference was less than 0.17. For dry catchments, additional information on the volume of streamflow, such as the annual mean flow or streamflow percentiles, were needed.

Even though the price for water level recorders has significantly gone down in recent years and their datalogging capacity has increased, it is not feasible to install a water level recorder in every ungauged catchment. It is, therefore, useful to also consider the use of other approaches to obtain water level data. Citizen science is now more frequently used to obtain environmental data over large areas (Savan *et al.*, 2003; Bonney *et al.*, 2009; Graham *et al.*, 2011; Fohringer *et al.*, 2015; Huddart *et al.*, 2016; Wiseman and Bardsley, 2016). Little *et al.* (2016) gave citizen scientists water level sounders to measure groundwater levels in private wells and found that these measurements provided valuable data on groundwater levels across a large area in Alberta, Canada, and that the measurements were relatively accurate; the root mean square error between citizen scientist observed water levels and pressure transducer based water levels ranged between 3 and 11 cm. Lowry and Fienen (2013) installed staff gauges in rivers and asked citizen scientists to send stream level measurements via text message. They showed that the accuracy of the crowd-sourced measurements and pressure transducer data were similar to the staff gauge gradations (root mean square error of 0.5 cm). However, it is not feasible to install a staff gauge in every ungauged catchment or to equip all citizen scientists with water level recorders. Therefore, it is useful to also design citizen science approaches that do not require staff gauges or water level sensors. Citizen scientists have for example successfully mapped the occurrence of streamflow in intermittent streams (Turner and Richter, 2011) and water levels are a standard measurement in the Earthwatch FreshWater Watch program (https://freshwaterwatch.thewaterhub.org/). Estimates of relative stream levels or stream level classes based on features in the stream or on the streambank (i.e., whether the water is above or below a certain rock) are easier and can be done more quickly than actual water level measurements and are, therefore, likely suitable for citizen science projects where no staff gauges are available. However, the (vertical) resolution of these data is less than those of actual stream level measurements.

Information from time lapse cameras or webcams can also be used to obtain information on stream water level classes. Pixel classification or image recognition to determine whether the water level is above or below a certain point can be used to determine the relative stream water level, even if no other information about the stream or the cross section is available. Several studies have shown that cameras can be used for accurate streamflow estimation (Muste *et al.*, 2011; Tsubaki *et al.*, 2011; Hilgersom and Luxemburg, 2012; Royem *et al.*, 2012; Stumpf *et al.*, 2016) but these studies used dedicated cameras that focused directly on the stream and often required information about the stream channel cross section. While promising, it is

unlikely that many of the ungauged streams will be equipped with these systems. However, streams are often included in the pictures of existing webcams and time lapse cameras that were installed for other reasons, e.g., to show the snow conditions on a ski-slope or to highlight the view from a hotel. The information from these webcams can be used to obtain information about the relative changes in the stream level or width but this information might not be very precise because of the sub-optimal angle of the camera. It is, thus, more likely that these images can be used to obtain information about the relative water level or stream width (class), rather than the actual water level. Remotely sensed satellite data can also be used to rank stream levels or stream width. These data, however, as promising as they are, have limitations regarding their accuracy and resolution (and will likely have so for the foreseeable future). Thus also for these measurements time series of level (or width) classes are more realistic than high-resolution time series of actual water levels.

For crowd-based (or citizen science) observations, but also for data from webcams or satellites, the resolution of the stream level data will be significantly poorer than for data obtained by a dedicated water level sensor. To determine the effect of this loss of information, we tested the usefulness of these new types of stream level class data for constraining a simple bucket-type hydrological model. The aim was to provide a basis for designing citizen science projects that collect data that are as informative as possible and that can be used to derive model-based streamflow time series. We pretended that stream level class observations were available continuously (daily) but only at a limited vertical resolution by transferring the streamflow data into stream level classes. We then tested how the number of stream level classes (i.e., the resolution) influenced the information content of the data with regard to constraining the model. Furthermore, we studied the effect of different locations of the class boundaries on model performance.

## 2. Methods

### 2.1 Study catchments and dataset

This study largely followed the methodology of Seibert and Vis (2016), who evaluated the value of water level time series for model calibration for almost 600 catchments in the contiguous US based on continuous, high-resolution stream level data. In this study the model was calibrated based on stream level class data for a subset of these catchments. The 100 catchments used in this study were chosen randomly from the catchments used by Seibert and Vis (2016) and are spread across the contiguous US. The hydrometric data for these 1 to 12584 km$^2$ catchments were obtained from the dataset for 671 catchments of Newman *et al.* (2015). The mean annual precipitation ($P$) was derived from DAYMET (Thornton et al., 2012) and varied for the different catchments between 249 and 3113 mm y$^{-1}$. The potential evapotranspiration ($E_{pot}$) was calculated with the Priestley–Taylor equation. The annual average runoff ratios calculated based on the precipitation at the mean elevation varied between 0.05 and 1.18 (between 0.12 to 0.93 for 90% of the catchments). The aridity index ($P/E_{pot}$) varied between 0.25 and 4.33. Of the 100 catchments, 22 are considered dry ($P/E_{pot} \leq 1.0$), 62 are considered humid ($1.0 < P/E_{pot} < 2.0$) and 16 catchments are considered wet ($P/E_{pot} \geq 2.0$).

## 2.2 Transformation of streamflow data into stream level classes

In order to determine how many stream level classes are needed for model calibration, the daily average streamflow data were converted into time series of $n$ stream level classes, where $n$ varied from 2 to 20 (Figure 1). In real citizen science projects the class boundaries are likely chosen based on features in the stream or on the stream bank (e.g. above or below a certain rock or

marker) but in this study we chose the boundaries so that each class contained the same number of data points. This meant that for the simulations with two classes we converted all streamflow values above the median to water level class 2 and all streamflow values below the median to water level class 1. Similarly, when using more classes we assigned the classes so that there were an equal number of measurements in each class (i.e. each class had observations for a fraction of $n^{-1}$ of the entire time period). For the cases with two and three stream level classes we also evaluated the optimal location of the class

boundaries. For this, we systematically varied the class boundaries by changing the fraction of the time that the water level was in each class.

## 2.3 Hydrological Model

The HBV (Hydroloiska Byråns Vattenavdelning) model (Bergström, 1992; Lindström et al., 1997) was used in the software implementation HBV-light (Seibert and Vis, 2012). The HBV model is a frequently used bucket-type model and consists of

different routines representing snow, soil, groundwater and stream routing processes. The HBV model, as it was applied here, has 14 free parameters, which are usually found by calibration or regionalisation. Elevation bands of 200 m were used to represent catchment topography, whereas only one lumped land-cover class was used for each catchment. The parameter ranges for the 14 model parameters in the HBV model were similar to those used by Seibert and Vis (2016) and represent the range of typical parameter values found in previous studies worldwide.

## 2.4 Model calibration and validation

For each catchment the HBV model was calibrated for the period 1.10.1982 - 30.9.1996 using a genetic optimization algorithm (Seibert, 2000). The data from the 1.1.1980 - 30.9.1982 period were used for warming up the model. For model calibration, we maximized the Spearman rank correlation coefficient ($r_s$; Spearman, 1904) between the stream level class data and the simulated streamflow. The Spearman rank correlation evaluates the dynamics of the modeled streamflow and is highest ($r_s = 1$)

when stream level and streamflow are monotonically related. The advantage of using the Spearman rank correlation for model calibration based on stream level class data is that no information about the rating curve is needed. While the rank correlation does not evaluate streamflow volumes and, thus, a value of one does not ensure a perfect fit, the rank correlation can still be beneficial for model calibration, especially in humid catchments, where flow is constrained by the water balance (Seibert and Vis, 2016). Here we used the rank correlation to evaluate the dynamics of the 'observed' stream level classes against the

simulated streamflow time series. One could argue that the use of class data leads to a large number of ties (measurements with the same (mean) rank for the water level class) and $r_s$ values of one can, due to these ties, thus, by definition of the

Spearman rank correlation, not be obtained. However, since we are not interested (or using) the absolute Spearman rank correlation values, but are only interested in the relative performance of different parameter sets, $r_s$ can still be used for model calibration because its value is highest when the dynamics of the stream level classes and streamflow are most similar.

For each catchment, we used 100 independent model calibration trials resulting in 100 parameter sets (one for each model calibration). For each of these (100) calibration trials, a total of 3500 model runs were done to find the optimum parameter set with the genetic algorithm. The 100 calibration parameter sets for each catchment were validated by comparing the simulated streamflow to the observed streamflow data using the model efficiency (Nash and Sutcliffe, 1970). For each catchment, the median value of the model efficiency for the 100 parameter sets was used to represent the performance of the model for that catchment.

## 2.5 Benchmarks

Different benchmarks were used to assess the performance of the models calibrated with the stream level class data: an upper benchmark that represents how good the model simulation would be if continuous streamflow data were available, and two lower benchmarks that represent a model simulation in the absence of any streamflow or stream level data.

For the upper benchmark ($R_{eff}$), the model was calibrated for each catchment using the streamflow data and optimizing the model efficiency (100 calibration trials per catchment, each consisting of 3500 model runs). The median model efficiency of these 100 calibration trials was used as the upper benchmark value for each catchment. Because the goal of this study was to assess the value of stream level class data for model calibration, rather than to evaluate the ability of the model to simulate the streamflow, all model validation results for the stream level class data are given as the difference in model efficiency relative to this upper benchmark ($\Delta R_{eff}$).

In addition, the simulations based on the stream level class data were also compared to the simulations based on calibrations derived from high-resolution stream level data ($r_{s\_\infty}$). Here the model was calibrated by optimizing the Spearman rank correlation between the observed and modeled streamflow (c.f. Seibert and Vis, 2016). These simulations represent a situation where a water level recorder is installed in the stream and this data is used for model calibration.

For the first lower benchmark ($L_{random}$), the model was run for each catchment 1000 times using randomly chosen parameters within the parameter ranges that were also used for model calibration. We used the median model performance for these 1000 parameter sets to represent the performance of the model with random parameters for that catchment. For the second lower benchmark ($L_{regional}$), the model was run 9900 times using the 100 calibrated parameter sets for each of the 99 other catchments and again the median model performance for these 9900 parameter sets was used to characterize the second lower benchmark for that catchment.

# 3. Results

## 3.1 Model performance as a function of the number of water level classes

Not surprisingly, the model efficiency was lower for the models calibrated with the stream level class data than for the models calibrated with the streamflow data (Figure 2 and Table 1). However, the differences between the models calibrated with the high-resolution stream level data and the models calibrated with water level class data was relatively small, as long as at least five stream level classes were used for model calibration (compare results for $r_{s\_5}$ and $r_{s\_\infty}$ in Figure 2 and Table 1). The median difference in efficiency for the models calibrated on high-resolution water level data and the models calibrated on five stream level classes was only 0.01. The median difference was 0.06 when only two stream level classes were used. These differences are small compared to the 0.17 difference in median model efficiency for the models calibrated on continuous streamflow ($R_{eff}$) and the high-resolution stream level data ($r_{s\_\infty}$).

A more detailed analysis of the increase in model performance with an increasing number of water level classes suggests that for the wet catchments model performance increased only slightly when increasing the number of water level classes from two to five but that for some of the dry catchments model performance increased significantly when using more than two water level classes (Figure 3). In general, the increase in model performance with an increasing number of stream level classes was largest for the catchments that have the largest difference in model performance between the upper and lower benchmarks (Figure 3).

## 3.2 Comparison with the benchmarks

Comparison of the performance of the models calibrated with stream level class data to the upper benchmark suggests that especially for the wet catchments the differences between traditional model calibration based on continuous streamflow data and the calibration based on the stream level class data were small (Figure 4a-b). For the dry catchments, model calibration based on stream level class data led to larger errors in the simulated streamflow (Figure 4a-b).

Comparison of the performance of the models calibrated with the stream level class data to the lower benchmarks suggests that the inclusion of stream level class data led to a huge improvement for some of the dry catchments (Figure 4c-d). However, the median improvement in model efficiency when using the data for two stream level classes compared to the lower benchmark ($L_{random}$) between the wet, humid and dry catchments was small (0.23, 0.23 and 0.15, respectively) and not statistically significant (Kruskal Wallace test p=0.09). The differences in the median improvement in the efficiency when using the data for five stream level classes compared to the lower benchmark ($L_{random}$) between the wet, humid and dry catchments were also small (0.23, 0.32 and 0.22, respectively) but statistically significant (Kruskal Wallis test p=0.02).

### 3.3 Optimal location of class boundaries

In order to determine the optimal location of the class boundaries, we systematically varied them for the cases with two and three water level classes. The results show that model performance generally improved when at least one class boundary was located at high stream levels. For example, for the case with two classes, the median model performance for the 100 catchments was highest when the class boundary was chosen so that the stream water level was in the lower class for 94% of the time and in the upper class for 6% of the time. The smallest median difference between the model performance for two classes and the upper benchmark occurred at the class boundary definition of 93-7% (Figure 5a). The variability in model performance also decreased when the boundary was chosen at a higher stream water level, so that for fewer catchments the difference between the median model performance (i.e., median performance of the 100 calibration parameter sets) and the upper benchmark was larger than 0.20 ($\Delta R_{eff}$ was larger than 0.20 for 86, 61, 22, and 22% of the catchments when the boundary was set at 10-90, 50-50, 90-10, and 94-6% of the time, respectively). There was no clear spatial pattern in the optimal location of the class boundaries and for a few catchments the optimal class boundary was located at a much lower stream level (Figure 5b). For the case with the three water level classes, on average for the 100 catchments, better model results were obtained when the boundary for the upper class was at a high water level, but the other boundary could either be at a high level or at a low level (Figure 6). Intermediate values for the lower boundary resulted in a poorer model performance. The median performance of the models calibrated with three water level classes for the 100 catchments was highest when the class boundaries were set so that the water level was in the lowest, medium and highest class 94, 5 and 1% of the time, respectively.

### 4. Discussion

### 4.1 Usefulness of stream level class data

The results of this study show that five stream level classes are as informative for model calibration as stream level data with a very high vertical resolution. This is good news for citizen science projects or webcam based analyses, as it is much easier to determine the stream level class when there are only a few classes than when there are many classes. The small difference between the performance of the models calibrated on data for a few stream level classes and the upper benchmark (Figure 4a-b) suggests that the stream level class data from citizen science approaches or webcam images is most useful for model calibration for wet catchments and that stream level class data for these catchments can be used in combination with a model to obtain time series of streamflow. This is encouraging, as it is likely much harder for citizen scientists to estimate streamflow than the stream level class and this way the streamflow data that are needed for water management or flood- or drought forecasting can be obtained from the stream level class data.

On the other hand, the large improvement of the models calibrated with stream level class data compared to the lower benchmark for some of the dry catchments (Figure 4c-d) suggests that stream level class data may be especially useful in improving model performance in some dry catchments when no other streamflow or stream level data are available. For these

catchments, the model performance of the lower benchmark (i.e. based on the random parameter sets) was very poor, while for the wet catchments the model performance of the lower benchmarks was already reasonably good (see color coding in Figure 3 and Figure 4). Thus the biggest gain in adding stream level class data was seen for the dry catchments, even though the absolute model performance was much poorer than for models calibrated on streamflow data. Seibert and Vis (2016)

showed that model calibration based on high-resolution stream level data worked best for wet catchments and that for dry catchments, additional data on the water balance were needed. Using such additional information may also improve model performance based on stream level class data for the dry catchments. What kind of additional information might be most useful in combination with stream level class data remains to be explored.

## 4.2 Location of the class boundaries

In practice, the boundaries between the different water level classes will be chosen based on features in the river or the stream bank that are easy to observe. The results from this study suggest that for most streams the optimal class boundaries should be located at the high flow levels, but not at the very highest flows. This high optimal class boundary is good news for model calibration based on opportunistic webcam images because high flows are usually easier to observe in these images than low flows because it may be difficult to see the water level at low flows when the camera does not focus directly on the stream.

Citizen scientists, on the other hand, are perhaps more likely to go out and estimate stream levels during nice weather conditions and low flow periods. However, people also tend to look at rivers when the water level is particularly high. The still relatively long time that the water level is in the highest class (e.g. 6% of the time or on average 22 days per year for the case with two water level classes for which the median model performance for the 100 catchments was highest) suggests that there is ample time for citizen scientist to observe the water levels during the high water level period. These results thus suggest that citizen

science projects should communicate to the participants that measurements during high water levels are important and worth collecting and transmitting.

The reasons that for the majority of the catchments the optimal boundary between the water level classes is located at high stream levels are related to the data, the model and the choice of the model evaluation criterion. The choice of a high water level class boundary helps to avoid the selection of a parameter set that leads to a too flashy streamflow response because the

water level is in the upper water level class for only a limited fraction of time. The information content of the water level class data, and thus its value for hydrological model calibration, is higher when we know that for some events the water level does not cross this boundary and for another set of events it does. If for every event the water level crosses the boundary because it is set at a low level, then it is not possible to distinguish between the responses of different events. Similarly, if the level is set too high, then the water level may cross the class boundary only a very few times so that no distinction can be made for the

response of the majority of the events. For the optimal boundary definition for the two classes at 94-6% of the time, there were on average between 2.2 and 27.2 switches between the two water level classes per year (median: 14.4; 25th and 75th percentile 8.0 and 17 respectively; Figure 7). One could also argue that the water level class data is most informative when the class boundaries are crossed as often as possible in the actual time series. For the majority of the catchments the water level class

boundary was most often crossed if it was set so that the water level was in the lower class for 60-80% of the time (Figure 7). For only eight of the 100 catchments the water level class boundary was most frequently crossed if it was set at such a level that it was in the lower class for less than 40 percent of the time; for eight other catchments the water level class boundary was crossed most often if the boundary was defined such that the water level was in the lower class for more than 80 percent of the time (Figure 7).

Wani et al. (2017) used censored data in a formal Baysian framework to simulate the combined sewer overflow in an urban catchment. Similar to the results here for the usefulness of two water level classes for model calibration, they show that binary data (i.e. a water level above or below a threshold) is very effective in reducing the parameter uncertainty in their rainfall runoff model. They show that the location of the threshold matters and highlight the high information content in crossing the threshold but also mention that it is difficult to determine the relation between the location of the threshold and the value of the data in reducing the parameter space because it depends on how close the system is to the threshold and how many times the threshold is exceeded.

The optimal location of the water level class boundaries is also dependent on the model validation criterion that is used. We used the model efficiency ($R_{eff}$, Nash and Sutcliffe, 1970) to evaluate model performance, which is known to give more weight to the evaluation of high flows (Krause et al., 2005; Schaefli and Gupta, 2007). A high water level class boundary provides more information for these high flows. Using a different model evaluation criterion that focuses less on the high flows would result in lower optimal class boundaries. For example, when using the log-transformed streamflow to evaluate the model performance, the model efficiency values (again median for the 100 catchments) were highest when the class boundary was chosen so that the stream water level was in the lower class for about 60% of the time when there are only two stream level classes, and the water level was in the lower, middle and upper class for about 10, 60, and 30% of the time when there were three water level classes. In other words, the exact location of the optimal water level class boundaries depends on the model evaluation criteria and should be chosen based on the objective of the study (e.g. simulation of the peaks, low flow periods or the water balance).

Because in real citizen science projects the boundaries will not be chosen based on optimality as discussed above, but will be chosen by citizens based on local conditions, such as identifiable features in the stream, this means that the usefulness of citizen science based water level class data for the simulation of different aspects of the hydrograph will differ. However, the investigation of theoretically optimal class boundaries is still valuable for at least two reasons. Firstly, these results can be used to provide guidance to citizen scientists on how to choose class boundaries, if at all possible. Secondly, such results can help to decide which citizen science based water level class data might be especially useful for the simulation of certain aspect of the hydrographs.

### 4.3 Limitations of this study when faced with the reality of citizen science based data collection

A challenge with citizen science-based stream level data is that observations are taken at irregular time intervals, with a limited vertical resolution and may contain errors. In this study, we addressed the issue of the limited vertical resolution by assessing

the value of stream level class data. More work is needed on the issue of irregular data to determine the number of observations that are needed and the best times of these observations. Model calibration using weekly stream level class data for the cases with two, three and five water level classes suggest that the deterioration in model performance when weekly data are used instead of daily data is very small. Previous studies on model calibration based on streamflow measurements have also

suggested that continuous streamflow data are not needed and only a few streamflow measurements, particularly during rainfall events, are already useful to constrain hydrological models because many of the streamflow measurements contain redundant information (Seibert and Beven, 2009; Rojas-Serna *et al.*, 2016).

In this study, we pretended to have stream level class data by transforming the streamflow data to stream level classes (Figure 1). This data, therefore, does not include any errors. In reality, citizen science data may contain errors and misclassification of

the water levels. The effects of data errors on model results needs to be tested as well. However, in this respect, it has to be mentioned that several studies have shown that citizen science data can be quite accurate (Cohn, 2008; Lowry and Fienen, 2013; Tye et al., 2016) (but not always, e.g. Savan et al. (2003)) and that traditional streamflow data also can have significant uncertainties and may even contain dis-informative information that affects model calibration (McMillan *et al.*, 2010; Beven and Westerberg, 2011).

**5. Conclusion**

This study demonstrates that stream level class data can be useful for calibrating hydrological models in otherwise ungauged catchments. The results confirm the conclusions from a previous study (Seibert and Vis, 2016) but more importantly extend the findings towards the use of stream level data for model calibration to cases where data is available at only a limited vertical resolution, such as in citizen science-based observation approaches or webcam image analysis. The results show that a small

number of stream level classes contain almost as much information for hydrological model calibration as high-resolution water level data. This is good news for citizen science approaches. We also found that class boundaries at high water levels result in the most informative water level class time series. While in practice the class boundaries are likely determined by the local situation (such as a rock that is covered by water at a certain level), the importance of high levels shows the value of motivating the public to collect data during high flow situations.

More generally, this study demonstrates how hydrological modeling can be used to evaluate the potential value of certain types of data. Similar approaches can be used to evaluate how much the information content of water level class data might decrease if observations are made at irregular times or with a certain amount of error. This information is crucial for the optimal design and implementation of citizen science-based observation approaches.

## Acknowledgements

We thank Andy Newman and Martyn Clark for making the data used in this study available. The ScienceCloud provided by S3IT at the University of Zurich enabled us to run the computational-intensive simulations on virtual machines. The comments of the two reviewers helped to clarify the text.

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

**Table 1 Median, maximum and minimum model efficiency for the 100 catchments for model calibrations using different types of data and the two lower benchmarks. Note that the difference in the median model efficiency for the model calibrations with all streamflow data ($R_{eff}$) and the median model efficiency for the model calibrations with data for *n* water level classes ($r_{s\_n}$) is not the same as the median of the differences in efficiency between the model calibrated with all streamflow data and the model calibrated with the stream level class data ($\Delta R_{eff}$) that is reported in the text and shown in the figures of the manuscript.**

| Data used for model calibration | | All catchments (n=100) | Dry catchments (n=22) | Humid catchments (n=62) | Wet catchments (n=16) |
|---|---|---|---|---|---|
| Streamflow data (upper benchmark, $R_{eff}$) | Median | 0.77* | 0.77 | 0.75 | 0.86 |
| | Max | 0.92 | 0.92 | 0.90 | 0.92 |
| | Min | 0.53 | 0.56 | 0.53 | 0.64 |
| Water level data ($r_{s\_\infty}$) | Median | 0.58 | 0.32 | 0.58 | 0.80 |
| | Max | 0.89 | 0.61 | 0.79 | 0.89 |
| | Min | -1.48 | -1.48 | 0.13 | 0.53 |
| 5 stream level classes ($r_{s\_5}$) | Median | 0.56 | 0.29 | 0.57 | 0.79 |
| | Max | 0.88 | 0.62 | 0.79 | 0.88 |
| | Min | -1.68 | -1.68 | 0.10 | 0.53 |
| 3 stream level classes ($r_{s\_3}$) | Median | 0.54 | 0.27 | 0.55 | 0.76 |
| | Max | 0.88 | 0.57 | 0.79 | 0.88 |
| | Min | -1.71 | -1.71 | -0.14 | 0.52 |
| 2 stream level classes ($r_{s\_2}$) | Median | 0.49 | 0.28 | 0.49 | 0.72 |
| | Max | 0.87 | 0.65 | 0.77 | 0.87 |
| | Min | -0.57 | -0.57 | -0.12 | 0.47 |
| Parameters from other catchments ($L_{regional}$) | Median | 0.43 | 0.21 | 0.43 | 0.70 |
| | Max | 0.79 | 0.50 | 0.65 | 0.79 |
| | Min | -5.56 | -5.56 | -2.54 | 0.43 |
| Random parameters ($L_{random}$) | Median | 0.25 | 0.11 | 0.26 | 0.56 |
| | Max | 0.76 | 0.38 | 0.66 | 0.76 |
| | Min | -6.04 | -6.04 | -1.60 | 0.13 |

* The median efficiency for the ~600 catchments studied by Seibert and Vis (2016) was 0.74

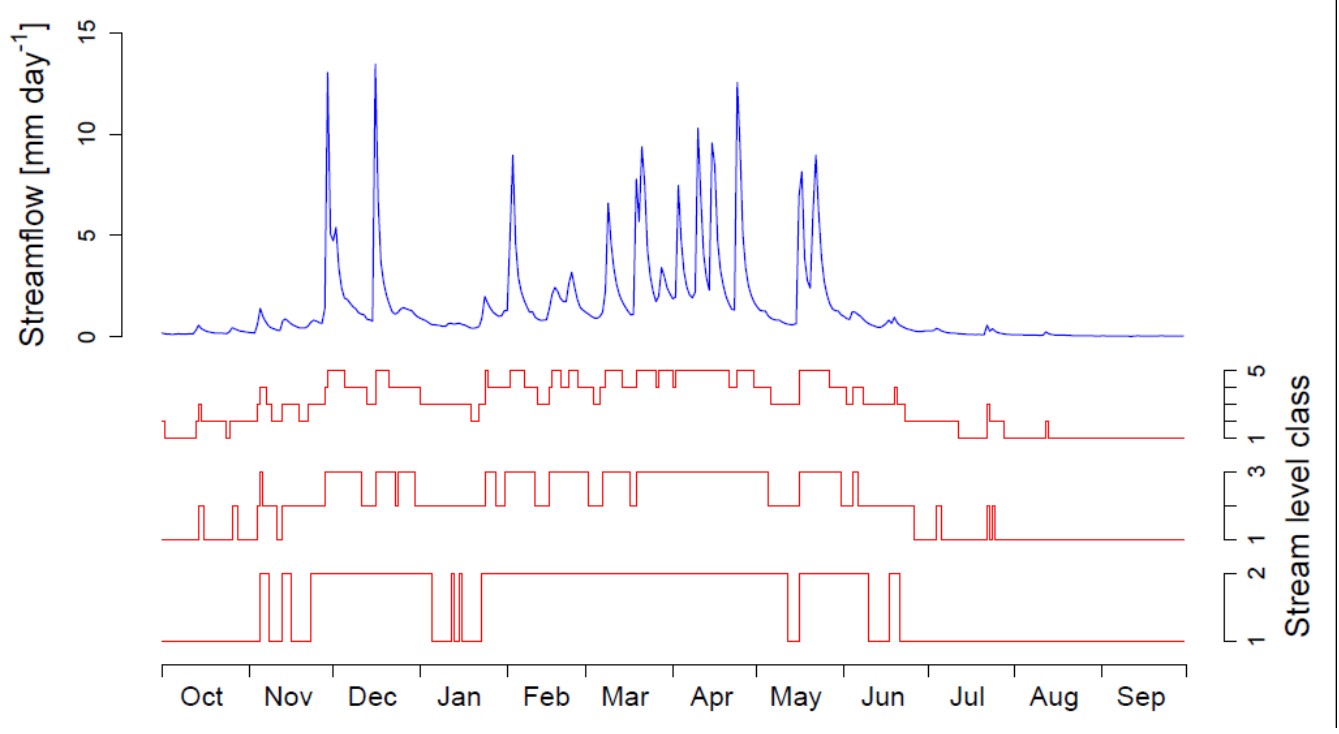

**Figure 1. Time series of the observed streamflow (blue) for the first year of simulation (October 1982 – September 1983) for the catchment 002011460 (Back Creek near Sunrise, VA, a medium sized catchment (235 km$^2$) with a medium aridity index (1.33) and the derived time series of stream level class for the case of two, three and five level classes (red), where the stream level is in each water level class for, respectively, 50%, 33% and 20% of the time.**

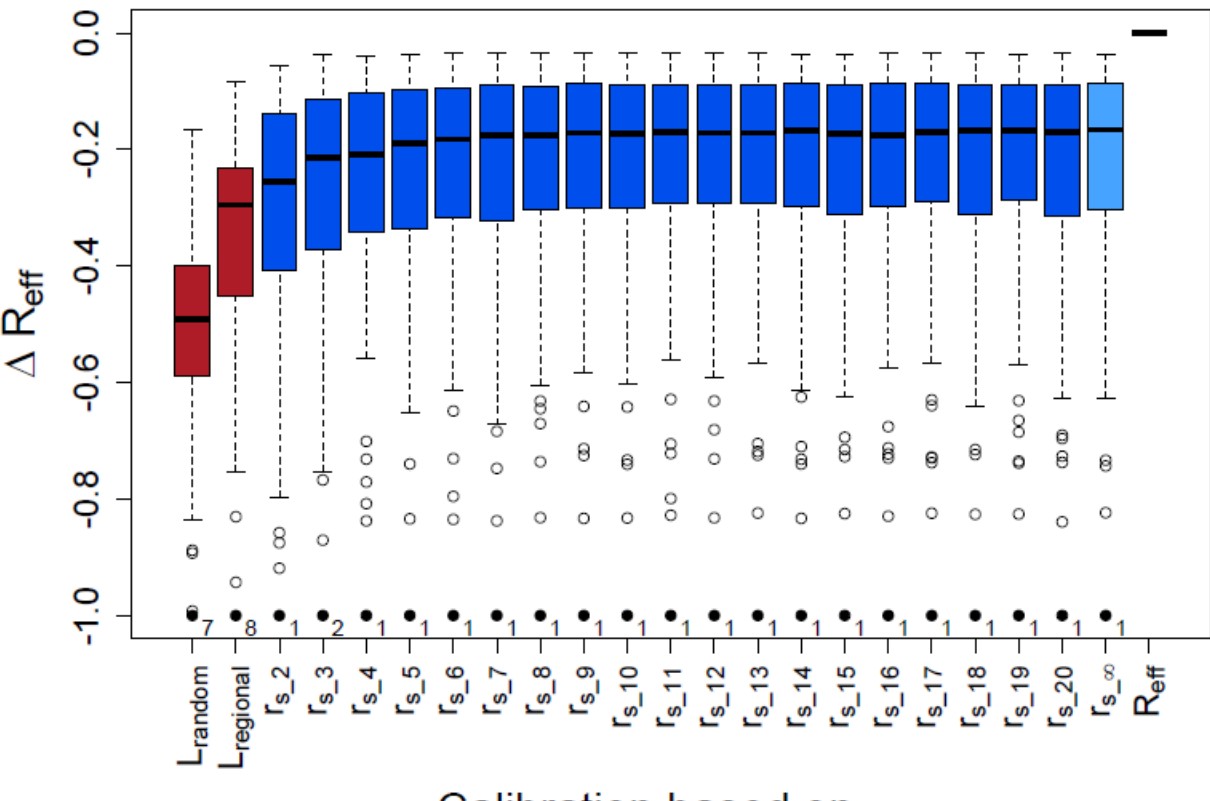

**Figure 2. Box plots of the difference in the median model efficiency and the upper benchmark ($\Delta R_{eff}$) for all 100 catchments for the models calibrated on stream level class data (2 to 20 classes; $r_{s\_n}$), models calibrated on high-resolution stream level data ($r_{s\_\infty}$), and the two lower benchmarks ($L_{random}$ and $L_{regional}$). The box represents the interquartile range, the solid line the median, the whiskers reach to the furthest catchment that is still within a distance of 1.5 times the interquartile range from the box and the dots represent the outliers.**

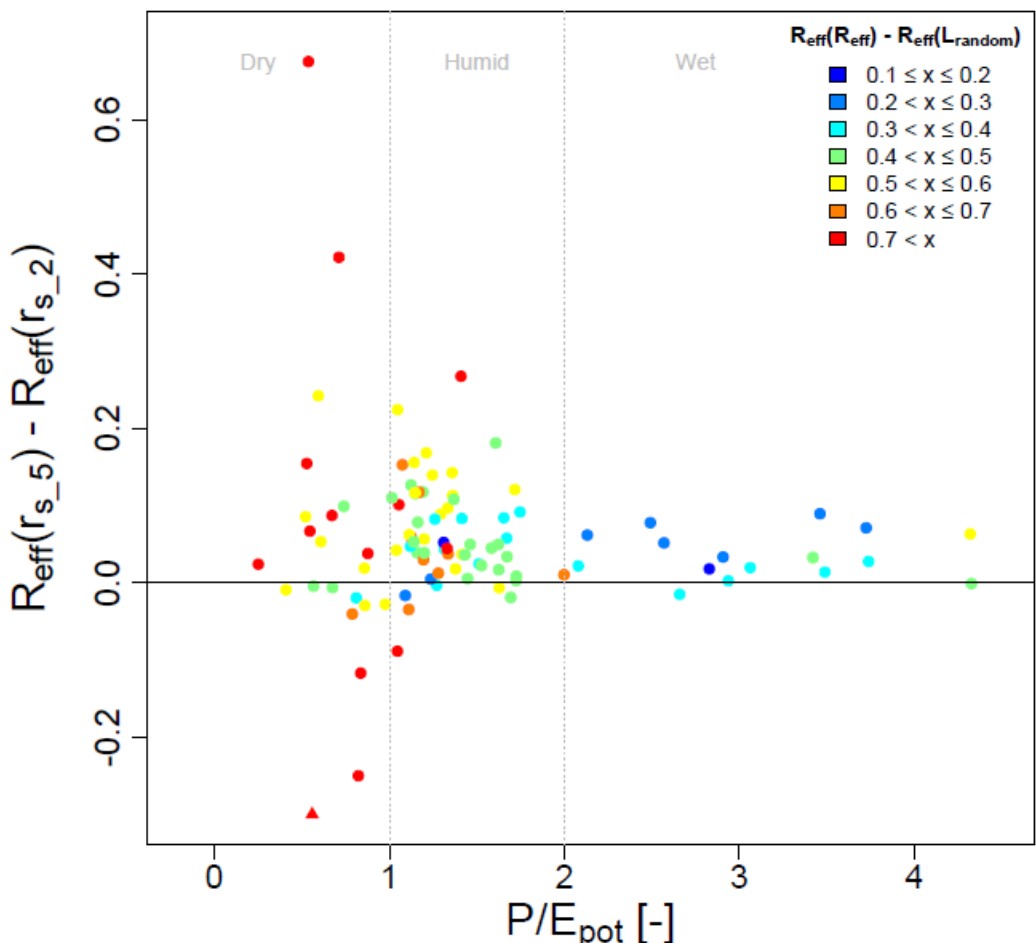

**Figure 3.** Difference in the median model validation result (model efficiency) for the models calibrated using two water level classes ($r_{s\_2}$) and five water level classes ($r_{s\_5}$) for all 100 catchments as a function of the aridity index ($P/E_{pot}$). The color of the symbols represents the difference between the upper and the lower benchmark (i.e. the difference in the median model performance when the model is calibrated with all available streamflow data ($R_{eff}$) and when the model is run with randomly selected parameters (i.e. without any calibration; $L_{random}$)). Triangles indicate outliers  that would plot outside the range of the y-axis.

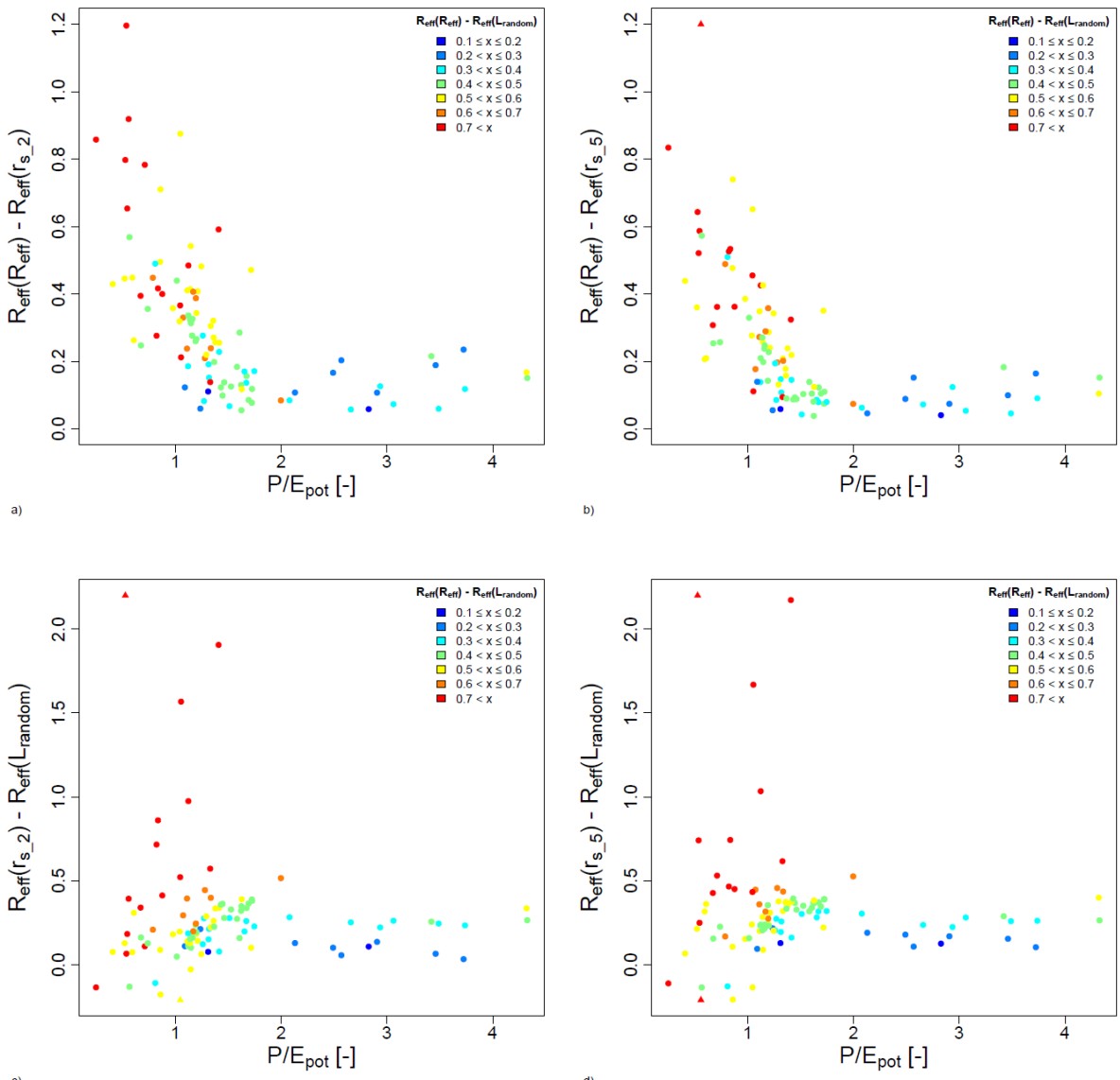

**Figure 4. Difference in model validation results (model efficiency) for the models calibrated with data from two ($r_{s\_2}$; left) and five ($r_{s\_5}$; right) stream level classes and the upper benchmark ($R_{eff}$) (upper row) and the lower benchmark ($L_{random}$; bottom row) as a function of the aridity index (P/E$_{pot}$). Each dot represents one catchment; the color of the symbol represents the difference in model efficiency between the upper and lower benchmark for that catchment. Note the difference in the scale of the y-axis for the comparison to the upper benchmark (upper row; a and b) and the lower benchmark (lower row; c and d). Triangles indicate outliers that would plot outside the range of the y-axis.**

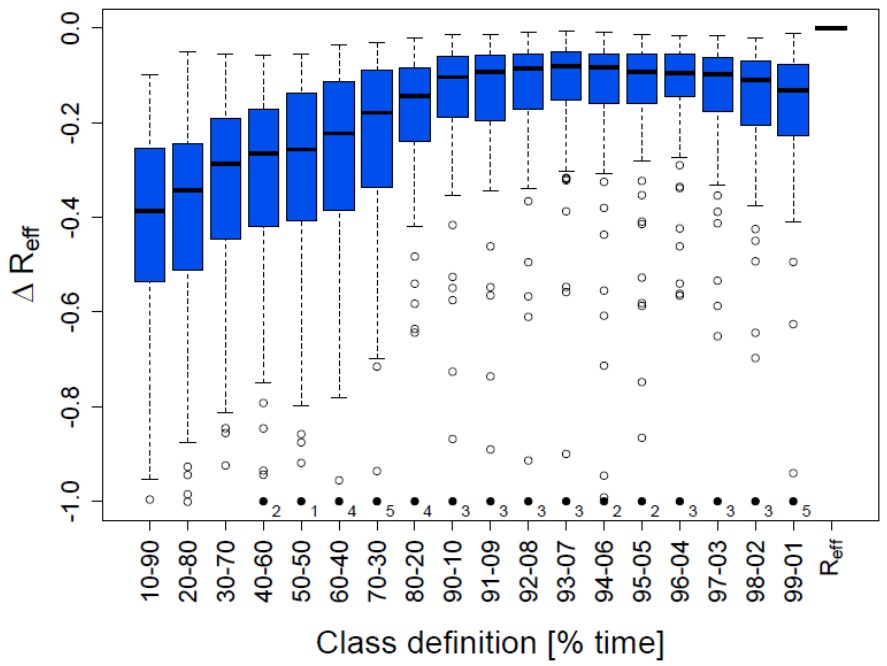

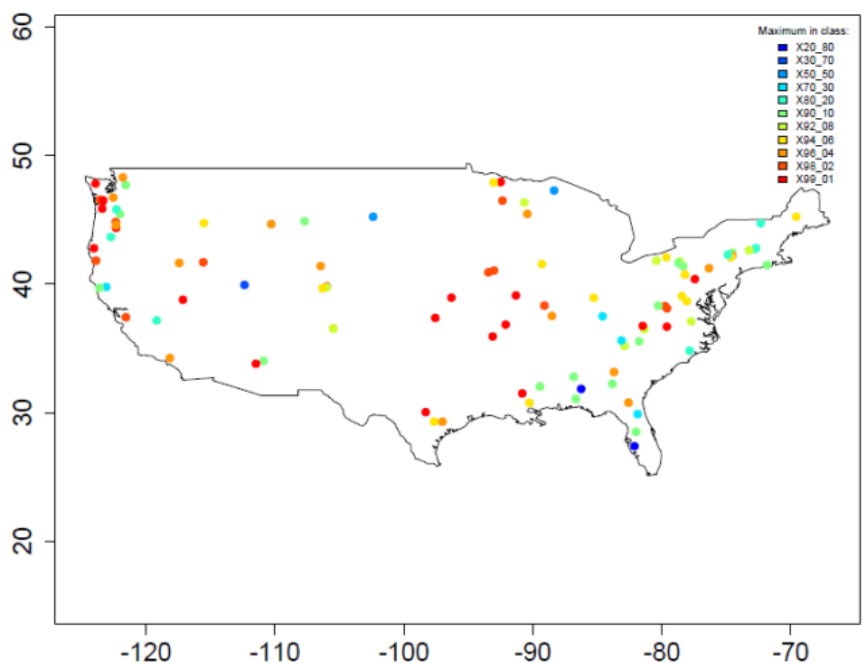

**Figure 5. Difference in median model validation results relative to the upper benchmark (*ΔR$_{eff}$*) for models calibrated with two water level classes for different class boundary definitions (a) and a map of the optimal class boundary definition for each catchment (b). As an example, 10-90 indicates that streamflow was in the lower water level class for 10% of the time and in the upper class for 90% of the time. The median difference in model efficiency is smallest when the class boundaries are set at 93-7%.**

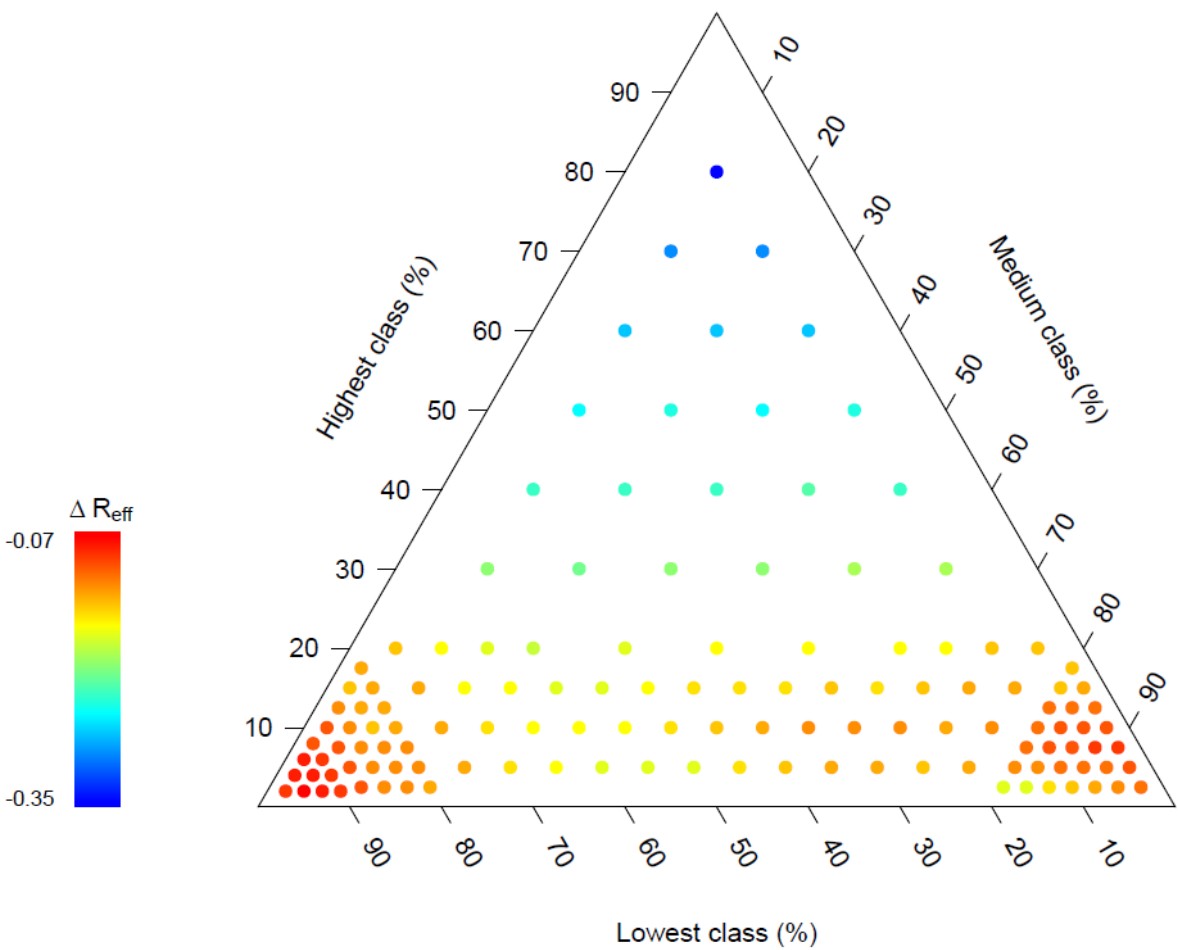

**Figure 6. Median difference in model efficiency (for models calibrated with data for three water level classes and the upper benchmark ($\Delta R_{eff}$) for different class boundaries.**

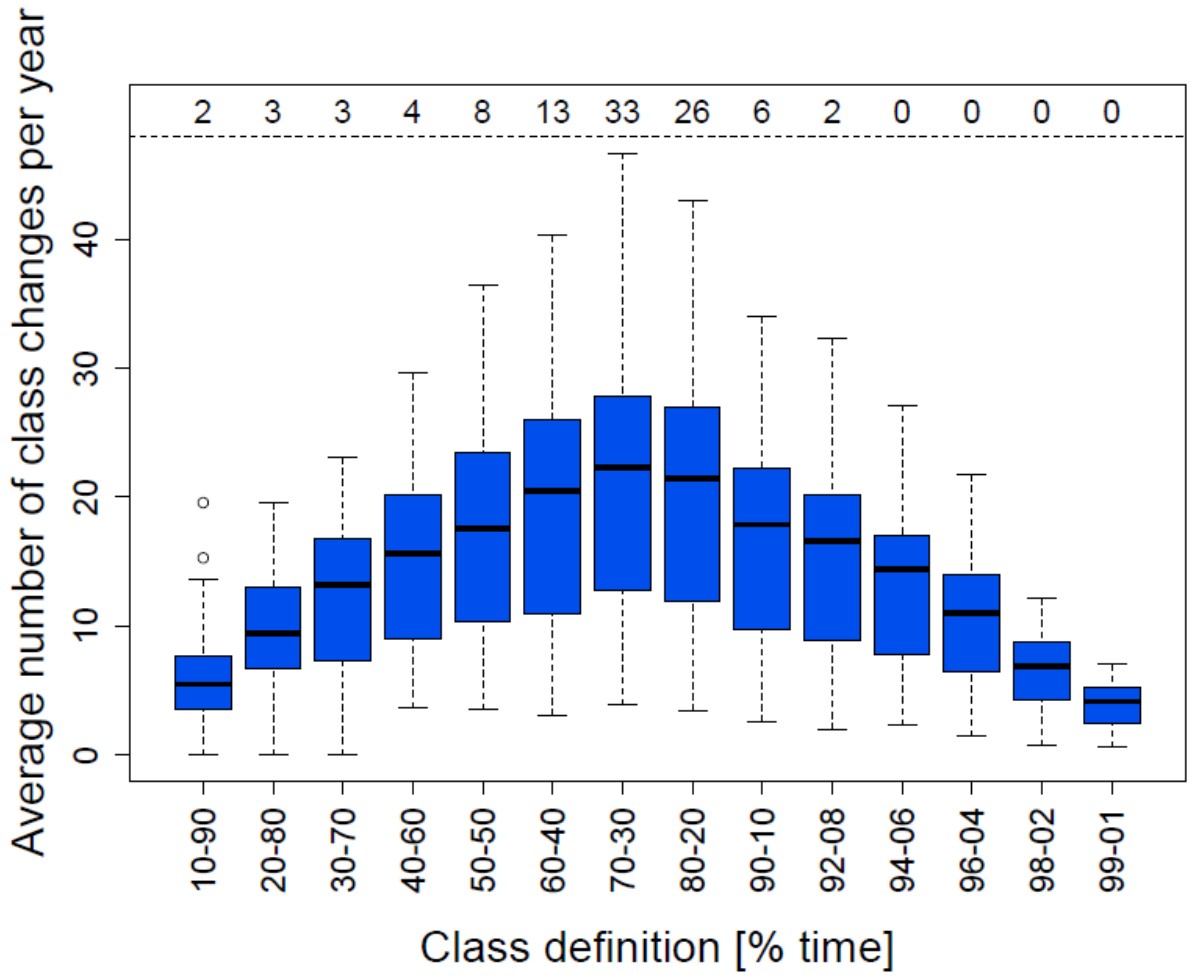

**Figure 7. Box plots of the average number of times per year that the water level switched from one class to another for different class definitions. In the top row the number of catchments for which the number of water level class switches was highest at that class definition. As an example, 80-20 indicates that streamflow was in the lower water level class for 80% of the time and in the upper water level class for 20% of the time, and for 26 of the 100 catchments this class boundary definition resulted in most class switches per year.**