# Peer review of "Information content of stream level class data for hydrological model calibration"

_Hydrology and Earth System Sciences, 2017_

## Referee Comment (RC1) · W. Buytaert (Referee) · 28 Apr 2017

This is a nice paper - very clearly written and overall well presented. The topic is novel and relevant - indeed I think that the insights are useful beyond citizen science, and help understanding the usefulness of other unconventional data sources such as cameras, and low-resolution sensors.

I have only a couple of concerns/queries:

1. The impact of measurement frequency on the performance of the models

The simulation of "citizen science" data pretends that stream level data are available at a daily level (p.3/16). This is a lot, and probably unrealistic for real citizen science applications. This matters, because the Nash Sutcliffe efficiency and many other performance measures are quite sensitive to timing errors, and daily measurements, even only of water level, will make it possible to calibrate the timing related parameters of a hydrological model (e.g. overland and channel flow velocities) pretty well for all but the smallest catchments. I expect that the constraining power of the data will decrease strongly if the frequency of measurement reduces. So it is a pity that this was not studied. Alternatively, it may be useful to evaluate the model performance using a measure that puts more weight on the water balance (e.g., bias), because this is of course the specific weakness of using water level data for calibration instead of streamflow data.

2. The reporting of the model efficiency.

The model efficiency measure R_eff is not defined (p.5/2). Only much further in the text, it is suggested that the Nash Sutcliffe efficiency is used (p.8/22-23). Is that correct? Irrespective of the definition of R_eff, I think that it would be useful to report the actual performance of the "upper benchmark", i.e. the models calibrated with streamflow data. This is useful to get an idea of the order of magnitude of model performance that can be obtained with the citizen science data (irrespective of the difference with a fully calibrated model).

3. Model calibration

The procedure used to calibrate the models is not clear to me. The manuscript states that "the model was calibrated 100 times, with each calibration trial consisting of 3500 model runs.", but I do not understand how exactly this is done. I suppose that the 3500 runs refer to different (sampled?) parameter sets, but what do the 100 times refer to? It suggests a kind of equifinality approach, but then I don't understand how this results in a single performance measures. Similarly, I don't understand how the 1000 randomly chosen parameters of the first lower benchmark (L_random), result in a single performance measure. I think that this needs to be clarified to make sure that it is reproducible, if only for confused minds like mine.

---

## Author Comment (AC1) · 10 May 2017

This is a nice paper - very clearly written and overall well presented. The topic is novel and relevant - indeed I think that the insights are useful beyond citizen science, and help understanding the usefulness of other unconventional data sources such as cameras, and low-resolution sensors.

> We thank the reviewer for the positive comments.

I have only a couple of concerns/queries:

1. The impact of measurement frequency on the performance of the models The simulation of "citizen science" data pretends that stream level data are available at a daily level (p.3/16). This is a lot, and probably unrealistic for real citizen science applications. This matters, because the Nash Sutcliffe efficiency and many other performance measures are quite sensitive to timing errors, and daily measurements, even only of water level, will make it possible to calibrate the timing related parameters of a hydrological model (e.g. overland and channel flow velocities) pretty well for all but the smallest catchments. I expect that the constraining power of the data will decrease strongly if the frequency of measurement reduces. So it is a pity that this was not studied. Alternatively, it may be useful to evaluate the model performance using a measure that puts more weight on the water balance (e.g., bias), because this is of course the specific weakness of using water level data for calibration instead of streamflow data.

> We agree that daily data is unlikely for citizen science projects (except perhaps in rare cases where there is a dedicated volunteer who takes daily measurements near his/her house). However, daily data is certainly likely for webcam or time-lapse camera images, which are usually renewed multiple times per hour.

> We already discussed these limitations in section 4.3, where we also mentioned that daily data contains a lot of redundant information and that previous studies have shown that a handful of measurements can be sufficient for model calibration (Rojas-Serna et al., 2016; Seibert and Beven, 2009). In response to this reviewer comment, we have now calibrated the model also with weekly (instead of daily) data. We then validated these parameterisations with the daily streamflow data. We did this for the case that weekly data are available for two, three and five stream level classes, stream levels and streamflow (Figure 1). The results show that the deterioration in model performance when weekly data are used instead of daily data is small, particularly for the stream level class data. We will include a description of these results in the revised version of the manuscript and if the editor thinks it is useful, can include the figure as well.

> To more realistically represent citizen science data we are working on a follow up study for Swiss catchments, where we will test the effects of different measurement intervals and the effect of the temporal distribution of the citizen science data on model calibration. Here, we will also include the effects of data errors. Because there are many possible scenarios to represent citizen science data, this leads to a very large number of simulations. We feel that it is too much to include all this information in this manuscript and that it would take the focus away from the central message that stream level class data are useful.

[Figure]

*Figure 1. Box plots of model performance relative to the upper benchmark ($R_{eff\_daily}$) for the the 100 catchments for model calibration with daily and weekly data for 2, 3, and 5 water level classes ($r_{s\_n}$), stream level data ($r_{s\_\infty}$) and weekly streamflow data ($R_{eff\_weekly}$). The results for the lower benchmarks are shown for comparison as well (note that for the lower benchmarks the model is not calibrated and there is thus no difference in model performance for daily and weekly data). The number of catchments for which the difference in model efficiency with the upper benchmark was >1 is given above the x-axis (indicated with the solid circles). The results for the lower benchmarks and the daily data are the same as those shown in Figure 1 in the manuscript.*

2. The reporting of the model efficiency. The model efficiency measure R_eff is not defined (p.5/2). Only much further in the text, it is suggested that the Nash Sutcliffe efficiency is used (p.8/22-23). Is that correct? Irrespective of the definition of R_eff, I think that it would be useful to report the actual performance of the "upper benchmark", i.e. the models calibrated with stream- flow data. This is useful to get an idea of the order of magnitude of model performance that can be obtained with the citizen science data (irrespective of the difference with a fully calibrated model).

We thank the reviewer for pointing out that we did not define model efficiency at the start of the manuscript. This is indeed the Nash Sutcliffe efficiency and we will make this clearer in the revised version of the manuscript.

We will also include a table with the minimum, maximum and median Nash Sutcliffe efficiencies for the upper benchmark ($R_{eff}$), the models calibration with high resolution water level data ($r_{s\_\infty}$), the models calibrated with two, three and five water level classes ($r_{s\_n}$) and the two lower benchmarks.

| Data used for model calibration | | All catchments (n=100) | Dry catchments (n=22) | Humid catchments (n=62) | Wet catchments (n=16) |
|---|---|---|---|---|---|
| Streamflow data (upper benchmark, $R_{eff}$) | Median | 0.77* | 0.77 | 0.75 | 0.86 |
| | Max | 0.92 | 0.92 | 0.90 | 0.92 |
| | Min | 0.53 | 0.56 | 0.53 | 0.64 |
| Water level data ($r_{s\_\infty}$) | Median | 0.58 | 0.32 | 0.58 | 0.80 |
| | Max | 0.89 | 0.61 | 0.79 | 0.89 |
| | Min | -1.48 | -1.48 | 0.13 | 0.53 |
| 5 stream level classes ($r_{s\_5}$) | Median | 0.56 | 0.29 | 0.57 | 0.79 |
| | Max | 0.88 | 0.62 | 0.79 | 0.88 |
| | Min | -1.68 | -1.68 | 0.10 | 0.53 |
| 3 stream level classes ($r_{s\_3}$) | Median | 0.54 | 0.27 | 0.55 | 0.76 |
| | Max | 0.88 | 0.57 | 0.79 | 0.88 |
| | Min | -1.71 | -1.71 | -0.14 | 0.52 |
| 2 stream level classes ($r_{s\_2}$) | Median | 0.49 | 0.28 | 0.49 | 0.72 |
| | Max | 0.87 | 0.65 | 0.77 | 0.87 |
| | Min | -0.57 | -0.57 | -0.12 | 0.47 |
| Parameters from other catchments ($L_{regional}$) | Median | 0.43 | 0.21 | 0.43 | 0.70 |
| | Max | 0.79 | 0.50 | 0.65 | 0.79 |
| | Min | -5.56 | -5.56 | -2.54 | 0.43 |
| Random parameters ($L_{random}$) | Median | 0.25 | 0.11 | 0.26 | 0.56 |
| | Max | 0.76 | 0.38 | 0.66 | 0.76 |
| | Min | -6.04 | -6.04 | -1.60 | 0.13 |

*For the 600+ catchments studied by Seibert and Vis (2016) the median efficiency was 0.74*

*Table 1. Median, maximum and minimum Nash Sutcliff efficiency for the 100 catchments for model calibrations using different types of data and the two lower benchmarks. Note that the difference in the median Nash Sutcliff efficiency for the model calibrations with all streamflow data ($R_{eff}$) and the median Nash Sutcliff efficiency for the model calibrations with data for n water level classes ($r_{s\_n}$) is not the same as the median of the differences in efficiency between the model calibrated with all streamflow data and the model calibrated with the stream level class data that is reported in the text and shown in the figures of the manuscript.*

3. Model calibration The procedure used to calibrate the models is not clear to me. The manuscript states that "the model was calibrated 100 times, with each calibration trial consisting of 3500 model runs.", but I do not understand how exactly this is done. I suppose that the 3500 runs refer to different (sampled?) parameter sets, but what do the 100 times refer to? It suggests a kind of equifinality approach, but then I don't understand how this results in a single performance measures. Similarly, I don't understand how the 1000 randomly chosen parameters of the first lower benchmark (L_random), result in a single performance measure. I think that this needs to be clarified to make sure that it is reproducible, if only for confused minds like mine.

We thank the reviewer for pointing out this unclarity and will improve the description in the next version of the manuscript. In short, we used 100 independent model calibration trials resulting in 100 parameter sets for each catchment (one for each model calibration). For each of these (100) calibration trials, a total of 3500 model runs were done to find the

optimum parameter set with the genetic algorithm. Thus, indeed 100 parameter sets were found for each dataset for each catchment. The median of the model performance for these 100 parameter sets is described in the text (and compared to the median performance of the model calibrated with the streamflow data).

For the lower benchmark, the 1000 random parameter sets result in 1000 model simulation results. We used the median model performance from these 1000 simulations to represent the performance of a model with random parameters.

---

## Referee Comment (RC2) · F. Nardi (Referee) · 22 May 2017

This research presents an investigation concerning the information content of stream level classes, potentially observed by citizens and/or using video-cameras, for improving hydrologic modelling performances in ungauged basins. The presented methodology and results show the potential value/capacity of informal hydrologic crowd-sourced observations - as respect to the case where/when high resolution flow monitoring or other standard hydrologic data are available - for monitoring and modelling river channel flows, especially in low contributing area river basins that are nowadays still lacking of adequate monitoring networks, also in developed regions.

The manuscript is well structured, presented and written and the subject/goals of the research, considering the actual importance of the active citizenship topic in hydrology (and not only), is of absolute interest for HESS. Nevertheless, there are some general, yet minor, issues and further few specific comments that I'm inserting hereafter that I strongly suggest authors to consider to improve the readability and clarity of the submitted work .

General Comments

GC.1) I fully agree with the first reviewer that the description of the calibration methodology is not clear. The performance parameter (Spearman rank), the modeling parameters used while performing the simulation used in the calibration process among others (see specific comments in the attached pdf) should be explained in more detail. The methodology description relies heavily on referenced works while the reader should be guided in independently following the manuscript without accessing other papers to understand data, methods and results.

GC.2) The characterization of the conversion of stream flow data into classes and the relationship of this crucial step with the stream flow level classification should be also better explained. The modeling results are presented only in the form of performance measures and this doesn't allow the reader in understanding the real "information content" of citizen-observed hydrologic monitoring data. Together with comments already introduced by first reviewer and already partially addressed by authors regarding the temporal sampling of flow data in both the monitoring and modeling process, I'd like to add a further major concern I have that is related to the quality/accuracy of the source (informal crowdsourced) data itself within the proposed research framework. To be more clear: results show that from 4-5 classes and up the modeling performances of the citizen-derived data are or may be "good", but in minor upstream rivers 5 classes of flow levels should be hard to be observed. While I approve the general concept and idea of the presented work I'd like to invite authors to express their view on the practical applications and related issues of the proposed method with specific regard to the issues of citizens gathering 5+ classes of flow level observations in upstream, often inaccessible, vegetation-dense creeks and very minor channels. In this regard
a sample picture from a real case study with a visual cross sections representing the potential analysis of the classes or a synthetic figure eventually associated with a flow chart to better depict the authors' view could constitute a solid improvement for this work.

GC.3) I understand authors are proposing a novel framework and testing the performances of flow level classes as calibration parameter for hydrologic models gathered from citizen science/data. And I assume the presented synthetic case study doesn't allow to dig into data, but I'd be glad to insert in the manuscript a river flow data/level plot comparing the different curves of hydrologic modeling results built upon the different monitoring datasets (highly detailed/resolution flow data vs citizen data ect). This would also help in addressing GC.1 for better describing the temporal/spatial sampling of parameters and results.

Specific/Minor comments

See attached PDF

Please also note the supplement to this comment:
http://www.hydrol-earth-syst-sci-discuss.net/hess-2017-72/hess-2017-72-RC2-supplement.pdf

**Supplement:**

[revised manuscript text omitted]

---

## Author Comment (AC2) · 31 May 2017

**Response to interactive review comment by F. Nardi (Referee)**

This research presents an investigation concerning the information content of stream level classes, potentially observed by citizens and/or using video-cameras, for improving hydrologic modelling performances in ungauged basins. The presented methodology and results show the potential value/capacity of informal hydrologic crowd-sourced observations - as respect to the case where/when high resolution flow monitoring or other standard hydrologic data are available - for monitoring and modelling river channel flows, especially in low contributing area river basins that are nowadays still lacking of adequate monitoring networks, also in developed regions.

The manuscript is well structured, presented and written and the subject/goals of the research, considering the actual importance of the active citizenship topic in hydrology (and not only), is of absolute interest for HESS. Nevertheless, there are some general, yet minor, issues and further few specific comments that I'm inserting hereafter that I strongly suggest authors to consider to improve the readability and clarity of the submitted work .

> We thank the reviewer for the positive assessment of our manuscript and appreciate the valuable comments on how to improve and clarify the text. Below we respond to the three general comments. Of course we will also consider all minor comments in the pdf when revising our manuscript.

*General Comments*

GC.1) I fully agree with the first reviewer that the description of the calibration methodology is not clear. The performance parameter (Spearman rank), the modeling parameters used while performing the simulation used in the calibration process among others (see specific comments in the attached pdf) should be explained in more detail.

The methodology description relies heavily on referenced works while the reader should be guided in independently following the manuscript without accessing other papers to understand data, methods and results.

> We will add more details on the use of the Spearman rank coefficient as an objective function and rewrite the methods section to better explain the methodology.

GC.2) The characterization of the conversion of stream flow data into classes and the relationship of this crucial step with the stream flow level classification should be also better explained.

> We will explain the conversion of the measured streamflow to the water level class data better. For the first simulations, with two classes, we converted all streamflow values above the median to water level class 2 and all streamflow values below the median to water level class 1. For all situations with more than two classes, we also assigned the classes so that there were an equal number of measurements for each class. See Figure 1 below for three examples of the time series of the measured streamflow and the assigned streamflow classes for the case that there were two, three, or five water level classes.

> Later (for Figures 4 and 5 of the manuscript), we changed the class boundaries for the case of two and three water level classes so that there were a different number of data points in each class.

The modeling results are presented only in the form of performance measures and this doesn't allow the reader in understanding the real "information content" of citizen-observed hydrologic monitoring data. Together with comments already introduced by first reviewer and already partially addressed by authors regarding the temporal sampling of flow data in both the monitoring and modeling process, I'd like to add a further major concern I have that is related to the quality/accuracy of the source (informal crowd-sourced) data itself within the proposed research framework. To be more clear: results show that from 4-5 classes and up the modeling performances of the citizen-derived data are or may be "good", but in minor upstream rivers 5 classes of flow levels should be hard to be observed. While I approve the general concept and idea of the presented work I'd like to invite authors to express their view on the practical applications and related issues of the proposed method with specific regard to the issues of citizens gathering 5+ classes of flow level observations in upstream, often inaccessible, vegetation-dense creeks and very minor channels. In this regard a sample picture from a real case study with a visual cross sections representing the potential analysis of the classes or a synthetic figure eventually associated with a flow chart to better depict the authors' view could constitute a solid improvement for this work.

> We agree that in some streams it might be difficult to distinguish five or more water level classes. However, our results show that already two or three classes can be informative and useful for model calibration and also that this is the case regardless of where the exact class boundaries are. While we talk about the water level being above or below a rock, in reality there are often multiple rocks that could be used to determine the water level class (see Figure 2 below). We describe in the manuscript (section 4.1) that it is good news for citizen science projects that two to three classes are already informative for model calibration because citizens are likely able to distinguish between two to three classes but not 20 classes. We will try to stress this point further in the revised manuscript.

> With regard to people's ability to observe stream level classes, we want to refer to the 'CrowdWater-Game' as a quick test and demonstration: https://docs.google.com/forms/d/e/1FAIpQLScJ_xYFeYRvBMZEEMoUI3BYddjhpSRRpnW0sty vFBJvqg8GTQ/viewform?c=0&w=1. This game includes photos of potential observation sites in Switzerland. The practical issues related to how to observe stream level classes are a central part of our CrowdWater project (see http://www.crowdwater.ch/).

> Also, please note that the smallest catchment that is included in our database is 1.2 km$^2$ and that these are thus well defined channels and not tiny headwater streams for which the water level may only rise a few cm.

GC.3) I understand authors are proposing a novel framework and testing the performances of flow level classes as calibration parameter for hydrologic models gathered from citizen science/data. And I assume the presented synthetic case study doesn't allow to dig into data, but I'd be glad to insert in the manuscript a river flow data/level plot comparing the different curves of hydrologic modeling results built upon the different monitoring datasets (highly detailed/resolution flow data vs citizen data ect). This would also help in addressing GC.1 for better describing the temporal/spatial sampling of parameters and results.

> We want to emphasize that we used real data from 100 catchments, the only synthetic aspect was that we generated stream level class data out of the real high-resolution time series (see Figure 1 below for an example). Given the number of catchments, it is difficult to

show time series (actually this would be difficult already for one catchment). Therefore, we argue that the summarizing assessment using model performance measures is more informative. However, we could include several plots showing observed and modeled responses for selected catchments in the supplementary material.

Specific/Minor comments

See attached PDF

We thank the reviewer for these comments and will address them in the revised version of the manuscript.

[Figure]

**Figure 1.** Time series of the observed streamflow (blue) for the first year of simulation (October 1982 – September 1983) for three median sized catchments (wet: 292 km$^2$; humid: 235 km$^2$; dry: 472 km$^2$) and the derived time series of stream level class for the case of two, three and five level classes (red), where the stream level is in each water level class for, respectively, 50%, 33% and 20% of the time. The inset shows the location of the three selected catchments. The aridity index of the catchments is 2.94 (humid), 1.33 (wet) and 0.71 (dry). Note that the scale for the observed streamflow is different for the three catchments.

[Figure]

**Figure 2.** Pictures of streams showing that multiple features can be used to define two to five stream level classes. For more pictures we refer to the CrowdWater game (https://docs.google.com/forms/d/e/1FAIpQLScJ_xYFeYRvBMZEEMoUI3BYddjhpSRRpnW0styvFBJvqg8GTQ/viewform?c=0&w =1).

---

## Author Response (AR1)

This is a nice paper - very clearly written and overall well presented. The topic is novel and relevant - indeed I think that the insights are useful beyond citizen science, and help understanding the usefulness of other unconventional data sources such as cameras, and low-resolution sensors.

**We thank the reviewer for the positive comments about the paper.**

I have only a couple of concerns/queries:

1. The impact of measurement frequency on the performance of the models The simulation of "citizen science" data pretends that stream level data are available at a daily level (p.3/16). This is a lot, and probably unrealistic for real citizen science applications. This matters, because the Nash Sutcliffe efficiency and many other performance measures are quite sensitive to timing errors, and daily measurements, even only of water level, will make it possible to calibrate the timing related parameters of a hydrological model (e.g. overland and channel flow velocities) pretty well for all but the smallest catchments. I expect that the constraining power of the data will decrease strongly if the frequency of measurement reduces. So it is a pity that this was not studied. Alternatively, it may be useful to evaluate the model performance using a measure that puts more weight on the water balance (e.g., bias), because this is of course the specific weakness of using water level data for calibration instead of streamflow data.

We discussed these limitations in section 4.3, where we also described that daily data contains a lot of redundant information and that previous studies have shown that only a handful of measurements may be sufficient for model calibration (Rojas-Serna et al., 2016; Seibert and Beven, 2009). We now include a statement that for the 2, 3, and 5 stream level classes the performance of the models calibrated with weekly (instead of daily) data and validated with the daily streamflow is very similar to the performance of the models that were calibrated with daily data (P10L1-8).

2. The reporting of the model efficiency. The model efficiency measure R\_eff is not defined (p.5/2). Only much further in the text, it is suggested that the Nash Sutcliffe efficiency is used (p.8/22-23). Is that correct? Irrespective of the definition of R\_eff, I think that it would be useful to report the actual performance of the "upper benchmark", i.e. the models calibrated with stream- flow data. This is useful to get an idea of the order of magnitude of model performance that can be obtained with the citizen science data (irrespective of the difference with a fully calibrated model).

We made it clearer throughout the text that this is the Nash Sutcliffe efficiency.

We also included a new table (Table 1) with the minimum, maximum and median Nash Sutcliffe efficiencies for the upper benchmark ( $R_{eff}$ ), the models calibration with high resolution water level data ( $r_s$ ), the models calibrated with 2, 3 and 5 water level classes and the two lower benchmarks.

3. Model calibration The procedure used to calibrate the models is not clear to me. The manuscript states that "the model was calibrated 100 times, with each calibration trial consisting of 3500 model runs.", but I do not understand how exactly this is done. I suppose that the 3500 runs refer to different (sampled?) parameter sets, but what do the 100 times refer to? It suggests a kind of

equifinality approach, but then I don't understand how this results in a single performance measures. Similarly, I don't understand how the 1000 randomly chosen parameters of the first lower benchmark (L\_random), result in a single performance measure. I think that this needs to be clarified to make sure that it is reproducible, if only for confused minds like mine.

We significantly rewrote the text to make this clearer (see section 2.4).

**Response to interactive review comment by F. Nardi (Referee)**

This research presents an investigation concerning the information content of stream level classes, potentially observed by citizens and/or using video-cameras, for improving hydrologic modelling performances in ungauged basins. The presented methodology and results show the potential value/capacity of informal hydrologic crowd-sourced observations - as respect to the case where/when high resolution flow monitoring or other standard hydrologic data are available - for monitoring and modelling river channel flows, especially in low contributing area river basins that are nowadays still lacking of adequate monitoring networks, also in developed regions.

The manuscript is well structured, presented and written and the subject/goals of the research, considering the actual importance of the active citizenship topic in hydrology (and not only), is of absolute interest for HESS. Nevertheless, there are some general, yet minor, issues and further few specific comments that I'm inserting hereafter that I strongly suggest authors to consider to improve the readability and clarity of the submitted work.

Below we respond to the three general comments. We have considered and adopted all minor comments in the pdf.

**General Comments**

GC.1) I fully agree with the first reviewer that the description of the calibration methodology is not clear. The performance parameter (Spearman rank), the modeling parameters used while performing the simulation used in the calibration process among others (see specific comments in the attached pdf) should be explained in more detail.

The methodology description relies heavily on referenced works while the reader should be guided in independently following the manuscript without accessing other papers to understand data, methods and results.

We added more details on the use of the Spearman rank coefficient as an objective function and rewrote the methods section to better explain the methodology (see section 2.4).

GC.2) The characterization of the conversion of stream flow data into classes and the relationship of this crucial step with the stream flow level classification should be also better explained.

We rewrote the section where we describe the conversion of the measured streamflow to the water level class data (see section 2.2) and included a new figure (Figure 1) to show this better.

The modeling results are presented only in the form of performance measures and this doesn't allow the reader in understanding the real "information content" of citizen-observed hydrologic monitoring data. Together with comments already introduced by first reviewer and already partially addressed by authors regarding the temporal sampling of flow data in both the monitoring and modeling process, I'd like to add a further major concern I have that is related to the quality/accuracy of the source (informal crowd-sourced) data itself within the proposed research framework. To be more clear: results show that from 4-5 classes and up the modeling performances of the citizen-derived data are or may be "good", but in minor upstream rivers 5 classes of flow levels should be hard to be observed. While I approve the general concept and idea of the presented work I'd like to invite authors to express their view on the practical applications and related issues of the proposed method with specific regard to the issues of citizens gathering 5+ classes of flow level observations in upstream, often inaccessible, vegetation-dense creeks and very minor channels. In this regard a sample picture from a real case study with a visual cross sections representing the potential analysis of the classes or a synthetic figure eventually associated with a flow chart to better depict the authors' view could constitute a solid improvement for this work.

We agree that in some streams it might be difficult to distinguish five or more water level classes. However, our results show that already two or three classes can be informative and useful for model calibration and also that this is the case regardless of where the exact class boundaries are. We carefully read through section 4.1 and think that we are clear in stating that it is good news for citizen science projects that two to three classes are already informative for model calibration because citizens are likely able to distinguish between two to three classes but not 20 classes.

GC.3) I understand authors are proposing a novel framework and testing the performances of flow level classes as calibration parameter for hydrologic models gathered from citizen science/data. And I assume the presented synthetic case study doesn't allow to dig into data, but I'd be glad to insert in the manuscript a river flow data/level plot comparing the different curves of hydrologic modeling results built upon the different monitoring datasets (highly detailed/resolution flow data vs citizen data ect). This would also help in addressing GC.1 for better describing the temporal/spatial sampling of parameters and results.

Given the number of catchments, it is difficult to show time series (actually this would be difficult already for one catchment). Therefore, we argue that the summarizing assessment using model performance measures is more informative (note that the new Table 1 gives all the summary information on the model efficiency).

In addition to these specific changes, we made several changes throughout the text to further clarify the text (see tracked changes).

[revised manuscript text omitted]